# Addressing Nutritional Knowledge Gaps in Inflammatory Bowel Disease: A Scoping Review

**DOI:** 10.3390/nu17050833

**Published:** 2025-02-27

**Authors:** Eleonora Ribaudi, Simone Amato, Guia Becherucci, Sara Carillo, Carlo Covello, Vincenzina Mora, Maria Chiara Mentella, Franco Scaldaferri, Antonio Gasbarrini, Caterina Fanali, Lucrezia Laterza, Daniele Napolitano

**Affiliations:** 1IBD Unit, UOC CEMAD Centro Malattie Dell’apparato Digerente, Dipartimento di Scienze Mediche e Chirurgiche Addominali ed Endocrino Metaboliche, Fondazione Policlinico Universitario Agostino Gemelli IRCCS, 00168 Rome, Italy; eleonora.ribaudi@policlinicogemelli.it (E.R.); guia.becherucci@guest.policlinicogemelli.it (G.B.); sara.carillo01@icatt.it (S.C.); covellocarlo@gmail.com (C.C.); vincenzina.mora@policlinicogemelli.it (V.M.); mariachiara.mentella@policlinicogemelli.it (M.C.M.); franco.scaldaferri@policlinicogemelli.it (F.S.); antonio.gasbarrini@unicatt.it (A.G.); caterina.fanali@policlinicogemelli.it (C.F.); lucrezia.laterza@policlinicogemelli.it (L.L.); daniele.napolitano@policlinicogemelli.it (D.N.); 2Cardiac Intensive Care Unit, Heart Transplant Centre and ECMO, Azienda Ospedaliera San Camillo Forlanini, 00152 Rome, Italy; 3Dipartimento di Medicina e Chirurgia Traslazionale, Università Cattolica del Sacro Cuore, 00168 Rome, Italy

**Keywords:** inflammatory bowel disease, Crohn’s disease, ulcerative colitis, diet, food quality, knowledge, food

## Abstract

This scoping review aims to map the existing literature on nutritional knowledge among people with IBD, identify gaps in current understanding, and provide guidance for future educational interventions. **Background**: Inflammatory bowel diseases (IBDs) are chronic conditions affecting the gastrointestinal tract, where nutrition plays a crucial role in symptom management. Despite its significance, patient knowledge about proper dietary practices remains limited, with widespread misconceptions potentially leading to suboptimal health outcomes. **Methods**: This review followed the Arksey and O’Malley framework and adhered to PRISMA 2020 guidelines. A systematic search was conducted in three databases (PubMed, Web of Science, and SCOPUS) for studies published between 2003 and 2024. Only studies involving adults (≥18 years) with IBD and focusing on nutritional knowledge were included. **Results**: From 1440 records initially identified, 23 studies met the inclusion criteria. The findings highlight that IBD patients often base dietary decisions on personal beliefs rather than evidence-based guidelines, leading to widespread food avoidance and increased risk of malnutrition. Misconceptions such as avoiding dairy, gluten, and fiber without professional advice were prevalent. Educational interventions, including personalized counseling and group sessions, showed the potential to improve nutritional knowledge and symptom management, though their application remains inconsistent across settings. **Conclusions**: IBD patients face significant gaps in nutritional knowledge, emphasizing the need for structured educational initiatives. A personalized, multidisciplinary approach, integrating dietary education into standard care, is essential to improve symptom control and enhance quality of life. Future research should focus on developing evidence-based interventions tailored to the unique needs of this population.

## 1. Introduction

Inflammatory bowel diseases (IBDs), including Crohn’s disease (CD) and ulcerative colitis (UC), are chronic inflammatory conditions affecting the gastrointestinal tract. Their precise etiology has been incompletely elucidated to date. However, research indicates that these diseases stem from an aberrant immune response to gut microbiota in genetically susceptible individuals [1]. This phenomenon often correlates with an imbalance in gut microbial communities and increased intestinal permeability, facilitating bacterial translocation and triggering inflammation [2].

The pivotal role of diet in modulating these processes is well recognized. Specifically, the Western dietary pattern has promoted intestinal barrier dysfunction and dysbiosis, increasing the predisposition to IBD [3,4,5]. Conversely, dietary patterns that promote a balanced microbiota and possess anti-inflammatory properties, such as the Mediterranean diet, play a vital role in protecting against the development of these conditions [3,6].

The current landscape of IBD treatment includes effective pharmaceutical interventions, with ongoing research continually increasing the range of available drug therapies. However, these treatments may not consistently achieve the desired therapeutic outcomes, and their efficacy may decrease over time [7,8]. Consequently, translational studies have underscored the potential of novel therapeutic concepts, such as the modulation of host–microbiome interactions, wherein diet assumes a fundamental role. Nutrition functions as a supplementary or alternative approach to pharmacological treatments, demonstrating effectiveness in regulating inflammation and managing symptoms associated with these diseases [9,10,11].

Recent research has highlighted a growing patient interest in the impact of diet and nutritional supplementation on the management of IBD. Patients increasingly recognize nutritional management as a pivotal element in controlling their symptoms [12,13,14,15,16]. Notably, a recent survey of adults diagnosed with IBD found that 59% of patients regarded diet to be at least as crucial, if not more so, than medication in addressing the disease. Moreover, 62% of patients reported successfully managing IBD symptoms through dietary modifications [16].

Food literacy is an emerging concept concerning the ability to gather, understand, process, and use relevant information in the food system [17]. Food literacy refers to the proficiency in food-related knowledge, skills, and behaviors that enable individuals to make informed, healthy, and culturally appropriate food choices [18]. This concept includes both critical knowledge (understanding the broader food system, including sustainability, culture, and emotions) and functional knowledge (practical skills for selecting, preparing, and consuming food) [18]. In the field of IBD, recent studies have revealed a widespread lack of understanding regarding optimal dietary choices and the variable effect of specific foods on gut health [9]. Frequently, patients are not provided with personalized dietary guidance and instead rely on general recommendations needing more scientific substantiation [9]. This dearth of precise information can result in suboptimal dietary selections, exacerbating symptoms and compromising overall wellbeing. Furthermore, such informational insufficiency often propels patients toward restrictive dietary regimens, increasing the likelihood of encountering nutritional deficiencies and malnourishment [16].

Comprehensive nutritional education empowers patients to make well-informed dietary choices, facilitating more efficacious disease management. Consequently, the integration of nutritional education should be regarded as a fundamental component in the comprehensive treatment of IBD, with the overarching goal of bolstering gut health and enhancing the holistic wellbeing of patients [16].

The aim of this scoping review is to map and summarize the current evidence regarding nutritional knowledge among IBD patients and highlight educational gaps. Through this review, we aim to provide a comprehensive overview that can guide future educational interventions and improve the clinical management of IBD through a more informed and aware nutritional approach.

## 2. Materials and Methods

### 2.1. Methodology

This scoping review was developed to provide a comprehensive overview of the level of nutritional knowledge of adult IBD patients. The methodology of this study is based on the framework outlined by Arksey and O’Malley [19] and according to the main recommendations.

This review was conducted following the PRISMA 2020 checklist to ensure transparency and completeness in reporting, which consists of four stages: searching, screening at the initial level, application of inclusion/exclusion criteria, and synthesis [20]. Since the aim was to describe the available literature without evaluating the included studies’ methodological quality, only the PRISMA checklist’s important elements were considered [21]. A detailed PRISMA checklist is included in the Appendix A.

### 2.2. Search Strategy

The search was conducted on three electronic databases: PubMed, Web of Science, and SCOPUS, using Boolean operators such as AND and OR. The database search strategy was refined using the following MeSH (Medical Object Headings) terms:

“Inflammatory Bowel Disease” OR “Colitis, Ulcerative”.

“Crohn Disease” AND “Food” OR “Food Quality” OR “Diet” AND “Knowledge”.

The final list of records was transferred to the Rayyan web-based review management system for study selection and screening [22]. During the initial screening, titles and abstracts were reviewed to exclude articles that did not meet the inclusion criteria; visible in Appendix A, this task was performed independently by three reviewers (ER; GB; SC) to ensure transparency and mitigate uncertainty in the review results. In case of disagreement during the study selection process, a fourth reviewer (D.N.) was consulted to resolve the dispute and finalize the list of included studies.

### 2.3. Eligibility Criteria

The search was restricted to articles published in English between 2003 and 2024. The target population consisted of adults aged 18 years and older diagnosed with IBD, specifically examining their levels of nutritional knowledge.

Titles and abstracts were initially screened, and studies without accessible abstracts or full texts were excluded. Narrative and systematic reviews were also excluded. For studies reporting mixed populations (adults and children), only those providing separate data for adult participants were considered eligible. The PRISMA flow diagram (Figure 1) illustrates the stepwise selection process including identification, screening, and inclusion phases [18]. Detailed characteristics of the included studies, such as authors, title, year of publication, study design, participant numbers and characteristics, assessment tools, reported nutritional knowledge, dietary behaviours, and key findings, are presented in Appendix A.

### 2.4. Inclusion Criteria

Studies involving adults (≥18 years) diagnosed with CD or UC.Research focusing on nutritional knowledge, dietary beliefs, or food-related quality of life.Articles published in peer-reviewed journals in English.Availability of full-text articles.

### 2.5. Exclusion Criteria

Studies focusing exclusively on pediatric populations.Research unrelated to nutritional knowledge or dietary behaviors in IBD.Articles lacking full-text access or written in languages other than English.

### 2.6. Risk of Bias Assessment

Given the exploratory nature of a scoping review, no formal risk-of-bias assessment was conducted. However, study limitations were noted where applicable.

### 2.7. Study Registration

This review was not registered, as registration is not always required for scoping review.

## 3. Results

### 3.1. Main Characteristics of the Included Studies

The literature search initially identified 1443 manuscripts. After removing duplicates and screening titles and abstracts against the inclusion criteria, 1062 articles were eligible for full-text review.

Following a detailed full-text screening, 1004 articles were excluded for not meeting the inclusion criteria (e.g., irrelevant population, outcomes not related to nutritional knowledge, or insufficient methodological rigor). Figure 1 presents a PRISMA flow diagram outlining the study selection process. The final selection included 23 studies deemed relevant to the research questions. Studies were included in the synthesis based on predefined eligibility criteria, ensuring relevance to the research question.

These studies consisted of 15 cross-sectional studies [13,14,15,16,23,24,25,26,27,28,29,30,31,32,33], 4 qualitative studies [34,35,36,37], 1 cohort study [38], 2 prospective studies [17,39], and 1 survey [40]. Figure 1 presents a PRISMA flow diagram outlining the study selection process.

An analysis of the geographic distribution revealed that the included studies were conducted across multiple continents: 14 in Europe [13,14,15,17,24,25,26,27,29,30,32,33,35,38], 4 in Asia [16,23,34,40], 4 in Oceania [28,31,35,37], and 1 in North America [36]. This geographic diversity reflects a broad range of contexts in understanding nutritional knowledge among people with IBD.

### 3.2. Food Preferences and Restrictions in Adult Patients IBD

Several studies have examined dietary behaviours and food choices among adults with IBD, highlighting a tendency to avoid foods perceived as harmful to the gastrointestinal tract [36]. Commonly avoided foods include dairy products [38,40], spicy foods [24,28,31,36,40], fatty foods [16,17,18,19,20,21,22,23,24,25,26,27,28,29,30,31,32,33,34,35,36,37,38,39,40], fibrous vegetables [14,15,16,17,18,19,20,21,22,23,24,25,26,27,28,29,30,31], alcohol [17,28,36], fried foods [28,40], and foods containing gluten [13,27]. Dietary modifications are often based on personal experiences or beliefs rather than evidence-based recommendations. For instance, Guida et al. (2021) [13] found that many patients adopted restrictive diets, such as lactose-free or gluten-free regimens, without professional guidance. Similarly, Godala et al. (2020) [24] reported that 85.4% of patients believed diet could trigger disease relapses, and 81.7% felt compelled to eliminate specific foods.

Marsilio et al. (2020) [33] noted that IBD patients showed slightly better nutritional knowledge than celiac disease patients but still struggled to meet dietary recommendations, particularly regarding nutrient source identification.

Studies also reveal that dietary habits improve after an IBD diagnosis in terms of meal frequency and food quality [26,28,29,34]. However, these changes are not always informed by accurate nutritional knowledge, leading to an increased risk of malnutrition and micronutrient deficiencies. Shafiee et al. (2020) [23] observed that patients with clinically active UC had inadequate dietary intake, particularly for macro- and micronutrients, compared to those in remission. Also, Walton and Alaunyte (2014) [27] and Fiorindi et al. (2022) [17] similarly found that lower intakes of carbohydrates and fiber were prevalent among IBD patients, often falling below national dietary recommendations. The impact of food avoidance is significant: Lim HS et al. (2018) [40] reported that micronutrient deficiencies, such as reduced intakes of calcium, zinc, and vitamin A, were more frequent in patients who excluded certain food groups, with a higher prevalence of malnutrition compared to those without dietary restrictions (*p* = 0.007).

These findings underscore the critical need for personalized nutritional management in IBD. Without professional dietary guidance, patients risk exacerbating nutritional deficiencies and compromising their overall health. Addressing these gaps with tailored interventions and evidence-based education can significantly improve symptom management and quality of life for individuals with IBD.

### 3.3. Nutritional Cognitive Interventions to Improve IBD Management

Cognitive nutritional interventions for adults with IBD focus on improving nutritional knowledge and encouraging evidence-based dietary choices to better manage disease symptoms, including strategies such as individual or group educational sessions, personalized nutritional counseling, and the use of written or digital informational resources. For instance, Crooks et al. (2022) [14] surveyed 223 healthcare professionals in the UK identifying key dietary components associated with IBD development and relapse and emphasizing the critical role of professional dietary guidance. The design and duration of interventions varied significantly: some programs offered weekly sessions over several months, while others delivered short-term intensive workshops. Czuber-Dochan et al. (2019) [35] and Sinclair et al. (2022) [30] stressed the need for tailored nutritional interventions to address the unique needs of IBD patients, highlighting their potential to enhance care and quality of life. For example, Sinclair et al. (2022) [38] identified discrepancies between patients’ beliefs and behaviors regarding dietary and supplementary practices, underscoring the importance of interventions that effectively translate knowledge into actionable dietary behaviors. Palamenghi et al. (2024) [32] demonstrated that patients with higher health engagement and better nutritional knowledge exhibited improved disease management and reduced relapse rates. Studies consistently demonstrate the benefits of cognitive nutritional interventions in improving nutritional knowledge and empowering patients to manage symptoms more effectively through diet. However, the success of these interventions relies heavily on professional guidance. A multidisciplinary approach that includes registered dietitians (RDs), gastroenterologists, and other healthcare professionals can provide the necessary support to implement effective and sustainable dietary changes, improving both symptom management and overall wellbeing [14,28,31,40].

### 3.4. Factors Influencing Nutritional Knowledge in IBD Patients

A range of factors influence the nutritional knowledge of adult with IBD, including formal education and access to accurate updated information, and personal experiences with the disease. Jimmy et al. (2016) [15] highlighted that dietary beliefs and behaviours are influenced by disease subtype, gender, age, and level of education, underlining the need for patients to receive dietary recommendations and individualized interventions to improve disease management and quality of life. Patients with a higher level of education generally demonstrate a better understanding of the nutritional implications of their condition.

The role of nutrition in IBD management is widely recognized. Vries et al. (2019) [16] and Czuber-Dochan et al. (2019) [35] found that most patients consider nutrition critical to managing their symptoms and acknowledge its profound psychosocial impact on food-related quality of life. However, dietary choices often have social consequences, such as limiting options when eating out or affecting social interactions, as reported by Zallot et al. (2012) [25] and others [25,26,35].

Day et al. (2021) [39] highlighted that previous surgical interventions, by reducing symptom severity, facilitate a more liberalized dietary approach, thereby enhancing food-related quality of life.

Personal experiences also play a significant role. Dietary tolerances are often learned through trial and error, with food choices adapted based on symptom patterns. Nowlin et al. (2020) [36] observed that dietary restrictions and modifications evolve over time, frequently influenced by the activity of the disease. Similarly, Jowett SL et al. (2003) [33] highlighted that individuals with UC perceive specific foods as either harmful or beneficial to their condition. Social support can further enhance nutritional knowledge. Support groups and networks offer a platform to share information and strategies, fostering a sense of community. However, a lack of reliable and accessible information sources often perpetuates misconceptions about diet, hindering effective management. Zallot et al. (2012) [25] noted that erroneous dietary beliefs could exacerbate the psychosocial burden of the disease.

These findings underscore the need for improved dietary education and evidence-based advice for individuals with IBD. Raising awareness of the integral role of diet in managing IBD can motivate the adoption of accurate information and better eating habits. Tailored educational interventions are essential to address these gaps, ultimately enhancing symptom management and quality of life [29].

## 4. Discussion

This paper aims to explore the level of nutrition knowledge among adult patients with IBD, focusing on identifying existing gaps and unmet educational needs. The literature review shows that in any complex disease, where diet is a contributing factor, it is difficult to understand the role of any single food, because dietary patterns involve exposure to different groups of foods. At present, the evidence on how diet influences IBD activity is insufficient, and the impact that food choices may have on the disease course remains unknown [41]. The studies analyzed [15,17,26,29] reveal that while there is increasing interest among patients regarding the role of diet in managing their condition, there remains significant variability in their understanding of optimal dietary choices and the impact of specific foods on intestinal health. This lack of knowledge often contributes to suboptimal dietary practices, including the adoption of restrictive diets that may temporarily alleviate symptoms but increase the risk of malnutrition and long-term nutritional deficiencies [41,42].

Micronutrient deficiencies are highly prevalent among IBD patients, affecting more than half of this population and resulting from reduced dietary intake, malabsorption due to intestinal inflammation, and increased nutrient losses [40]. The most reported deficiencies include iron, vitamin B12, vitamin D, vitamin K, folic acid, selenium, zinc, vitamin B6, and vitamin B1, with higher prevalence in CD compared to UC and more pronounced during active disease phases [41]. Iron deficiency, the leading cause of anemia in IBD, often arises from chronic inflammation and gastrointestinal blood loss. Similarly, vitamin D and vitamin K deficiencies correlate with heightened inflammatory states, while vitamin B12 deficiency may go undiagnosed when relying solely on serum levels [43]. Addressing these deficiencies is crucial for optimizing clinical outcomes and preventing complications, such as impaired wound healing, increased infection risk, and bone health deterioration. Early detection through routine monitoring and targeted supplementation can mitigate the risk of irreversible sequelae. Current guidelines advocate for regular micronutrient screening in IBD patients, particularly during disease flares, post-intestinal resection, and in those adhering to restrictive diets. A multidisciplinary approach involving gastroenterologists, RDs, and primary care providers is essential to ensure timely diagnosis and appropriate treatment, ultimately enhancing patient outcomes and quality of life [40,42].

One of the key themes that emerged from the literature is the tendency of IBD patients to avoid certain foods perceived as irritants or harmful to their condition. These include dairy products [38,40], spicy foods [24,28,31,36,40], fatty foods [16,40], fibrous vegetables [14,36], alcohol [17,28,36], fried foods [28,40], and gluten [13,27]. These dietary behaviors are frequently based on personal experiences and individual beliefs rather than evidence-based dietary recommendations. This highlights the importance of addressing these misconceptions through structured educational interventions that are tailored to the cultural and personal contexts of the patients. This finding is in line with previous studies that have highlighted how many patients with IBD tend to modify their diet after diagnosis, often without receiving adequate nutritional counseling [14,17,24,27,28,34,36,38,40].

The literature review [44] shows a wide variation in the cognitive and educational interventions offered to patients regarding nutrition, ranging from individual educational sessions to workshops group, personalized nutritional counseling [16,26], and providing informational materials [16]. Although these interventions have demonstrated some success in improving patients’ nutritional knowledge and their ability to manage symptoms through diet, the absence of a standardized, coordinated approach between healthcare professionals and patients remains a significant challenge [16,26]. It is crucial to align these interventions with existing guidelines, such as the ECCO and ESPEN guidelines, which emphasize the role of a multidisciplinary team in managing IBD. This alignment would ensure a more structured and unified approach to patient education [28,36,45]. There is a clear need for the development of solid evidence-based dietary guidelines and the implementation of educational programs tailored to the individual needs of patients [16,26,39].

Another crucial fact is the influence of formal education on the nutrition knowledge of IBD patients [16,17,29,35]. Patients with higher levels of education tend to have a better understanding of the nutritional implications of their condition, suggesting that access to accurate and updated information plays a crucial role in shaping dietary choices and, ultimately, in improving disease management [35,38]. However, even among more educated patients, misinformation and confusion about appropriate dietary practices persist, underlining the need for clearer and more consistent messaging from healthcare providers [32,36]. Additionally, healthcare providers should receive training to address these misconceptions effectively, ensuring that the information shared with patients is consistent, evidence-based, and practical for daily life. There is a growing recognition of the need for a multidisciplinary approach to the management of IBD, which incorporates gastroenterologists, RDs, nurses, and other healthcare professionals who can provide ongoing nutritional support [14,28,31,40]. Such an approach would help patients implement effective and sustainable dietary changes, allowing for a more holistic view of patient care, addressing the physical aspects of the disease and the psychosocial factors that influence dietary behaviours and overall wellbeing [14,28,31,40].

The implications for clinical practice are significant: providing evidence-based, personalized, and continuous dietary counseling is essential to improve the quality of life for patients with IBD and to optimize symptom control. Diet and nutrition should be integrated more fully into the standard care protocols for IBD, with clear, accessible guidelines for patients and healthcare providers. In particular, the inclusion of nutritionists in routine care and the establishment of clear referral pathways can bridge the gap between patients’ needs and the services provided. Furthermore, more robust research is needed to develop targeted nutritional interventions that can be adapted to different stages of the disease, considering the unique dietary needs of individuals with IBD [42,44,45].

Several dietary approaches have shown promise in IBD management. The Mediterranean diet, characterized by high intake of fresh fruits and vegetables, monounsaturated fats, complex carbohydrates, and lean proteins, alongside limited ultra-processed foods, added sugars, and salt, is broadly recommended for IBD patients unless contraindicated [45]. While no specific diet has consistently reduced flare rates, a low red and processed meat diet may lower UC flares without similar benefits for CD. Adherence to the Mediterranean diet has been associated with lower disease activity, reduced inflammatory biomarkers, and improved quality of life in both CD and UC patients [46]. Exclusive enteral nutrition (EEN), utilizing liquid nutrition formulations, has demonstrated efficacy in inducing clinical remission and endoscopic response in CD exclusion diet (CDED), a form of partial enteral nutrition (PEN), represents a viable alternative for patients with mild to moderate CD, offering comparable effectiveness to EEN while enhancing treatment adherence. EEN is also recommended as preoperative therapy for malnourished CD to optimize nutritional status and reduce postoperative complications [42,44,47].

The low-FODMAP diet has shown promise in providing short-term relief of gastrointestinal symptoms in IBD patients with coexisting IBS-like symptoms. However, its potential negative impact on the gut microbiome, including reduced beneficial bacteria and decreased butyrate production, discourages long-term use [48]. Instead, short-term application during symptomatic flares should be followed by a transition to a balanced Mediterranean-style diet for long-term management. Studies by Więcek et al. (2022) [46], Cox et al. (2020) [49], and Pedersen et al. (2017) [50] demonstrated significant symptom relief and quality-of-life improvements among IBD patients following low-FODMAP interventions, though caution is warranted regarding prolonged adherence due to potential nutritional deficiencies and gut microbiome alterations. These findings underscore the importance of dietitian supervision to ensure dietary adequacy and avoid unintended consequences [46,49,50].

RDs play a pivotal role in the multidisciplinary management of IBD, providing essential support in nutritional assessment, individualized counseling, and the management of complex nutritional therapies, including parenteral and enteral nutrition. It is recommended that all newly diagnosed IBD patients have access to an RD to facilitate tailored educational interventions, improve nutritional literacy, and enhance patient outcomes. Routine screening for malnutrition remains vital, with a focus on signs such as unintended weight loss, edema, fluid retention, and loss of fat and muscle mass. Common nutritional deficiencies in IBD include vitamin D, iron, and vitamin B12, with additional risks for zinc, copper, fat-soluble vitamins, and folic acid deficiencies, particularly among patients on methotrexate and sulfasalazine. Early detection and targeted interventions are essential for addressing these deficiencies and improving overall patient care [8,10,17,51].

Dietary practices are deeply shaped by cultural beliefs, which can significantly influence nutritional knowledge and behaviors among IBD patients. Recent studies have highlighted how cultural background affects patients’ perceptions of dietary recommendations, adherence to specific diets, and overall understanding of the role of nutrition in disease management [51]. Furthermore, the rise of short-video sharing platforms has facilitated access to dietary advice; however, the quality and reliability of the information remain inconsistent. A recent study evaluating IBD diet-related videos on popular Chinese platforms revealed that content provided by medical professionals (particularly RDs and gastroenterologists) was significantly more comprehensive and reliable compared to non-medical sources. These findings underscore the importance of promoting high-quality, culturally sensitive educational resources to improve nutritional literacy among IBD patients [51].

The main limitations of this scoping review include the variability in the design and rigor of the included studies, and the exclusion of non-English publications, which may limit the generalizability of the findings. Furthermore, most studies focus on adult patients, excluding children and adolescents, limiting the understanding of the needs of these populations. Future studies should prioritize a broader inclusion of patient populations, aiming to establish standardized methodologies to improve the comparability of findings across different studies, while also including younger groups to develop more appropriate management strategies.

## 5. Conclusions

In conclusion, this paper highlights the ongoing need for targeted nutritional education for patients with IBD, emphasizing the importance of evidence-based dietary guidance to improve disease management and quality of life. While patients show a growing interest in the role of diet, significant gaps persist in their understanding of optimal food choices, often leading to restrictive practices that increase the risk of malnutrition.

Micronutrient deficiencies remain prevalent, especially during active disease phases and in CD. Early detection through routine monitoring and targeted supplementation is crucial for preventing complications and improving outcomes.

Structured, personalized dietary interventions, such as the Mediterranean diet and short-term low-FODMAP applications, have shown promise in symptom management, while PEN, EEN, and CDED offer effective options for CD. The involvement of RDs in multidisciplinary care is essential to bridge current knowledge gaps, prevent unnecessary restrictions, and promote sustainable dietary practices.

Future research should focus on the long-term impact of dietary interventions, the role of specific dietary patterns, and the cultural factors influencing dietary choices. Integrating structured nutritional counseling into routine IBD care can significantly enhance patient outcomes and overall wellbeing.

## Figures and Tables

**Figure 1 nutrients-17-00833-f001:**
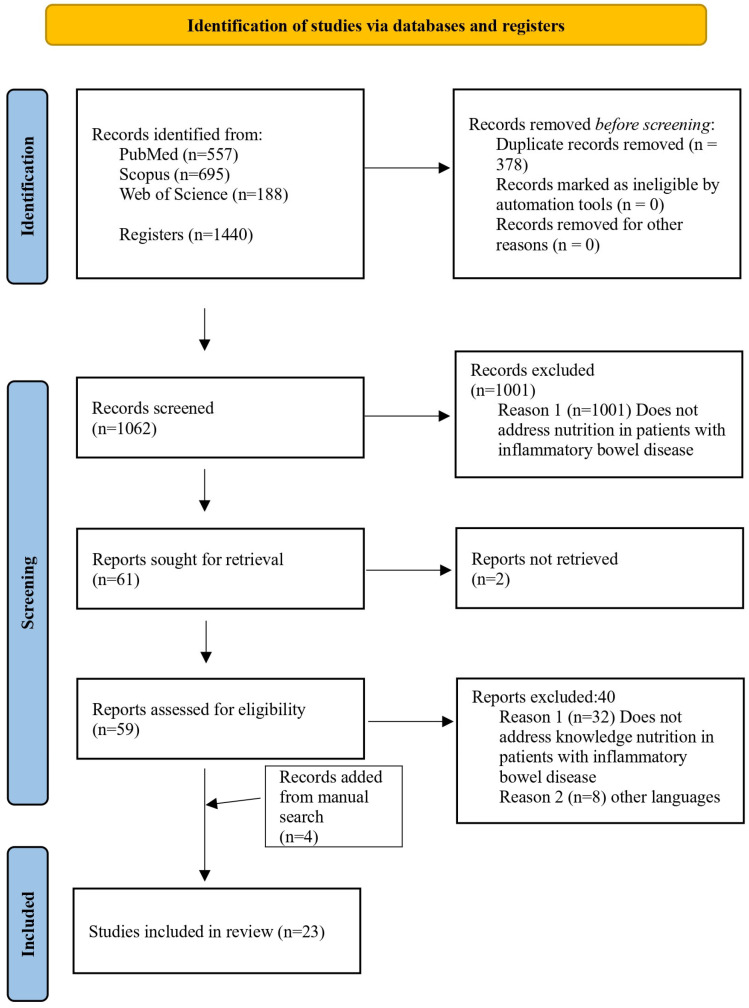
Flowchart for studies included in the scoping review.

## Data Availability

Data available in a publicly accessible repository. The data presented in this study are openly available in PubMed.

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
