# Peer review of "Addressing Nutritional Knowledge Gaps in Inflammatory Bowel Disease: A Scoping Review"

_nutrients, 2025, doi:10.3390/nu17050833_

Round 1
Reviewer 1 Report
Comments and Suggestions for Authors
This scoping review examines the current state of nutritional knowledge among patients with inflammatory bowel disease (IBD), highlighting significant gaps and misconceptions. The authors provide a well-balanced analysis of various dietary approaches and their impact on disease management, emphasizing the role of misinformation in shaping patient behavior. The review effectively underscores the importance of structured educational interventions to improve nutritional literacy and, ultimately, patient outcomes.
The authors deserve credit for addressing a real and growing issue in clinical practice: the rise of "dietary anxiety"—or sitophobia—among IBD patients, often fueled by social media and self-proclaimed nutrition experts. Many patients adopt restrictive diets based on unverified claims, leading to unnecessary food avoidance and potential malnutrition. The review does an excellent job of contextualizing these issues with evidence from multiple dietary patterns, including Mediterranean, vegetarian, and low-carbohydrate diets. However, one area that requires further exploration is the role of the low FODMAP diet. This approach has been shown to significantly reduce functional gastrointestinal symptoms in IBD, particularly during disease flares, and should be included as a viable dietary strategy rather than just being lumped in with broad food avoidance behaviors.
Another critical omission is the discussion of food-based medical therapies, such as exclusive enteral nutrition (EEN) with formulations like Modulen or the elemental diet for Crohn’s disease. These are well-established interventions, particularly in pediatric populations, but also have applications in adults who require nutritional optimization without exacerbating inflammation. Given the review's focus on knowledge gaps, it would be beneficial to assess whether patients are aware of these therapeutic options and their clinical efficacy.
Overall, this is a well-researched and thoughtful review that highlights an often-overlooked aspect of IBD management. Addressing the above points would further strengthen its clinical relevance and applicability.
Author Response
Comment 1:
The review effectively underscores the importance of structured educational interventions to improve nutritional literacy and, ultimately, patient outcomes. However, one area that requires further exploration is the role of the low FODMAP diet. This approach has been shown to significantly reduce functional gastrointestinal symptoms in IBD, particularly during disease flares, and should be included as a viable dietary strategy rather than just being lumped in with broad food avoidance behaviors.
Response 1:
Thank you for your insightful comment. We agree that the low FODMAP diet represents a valuable dietary strategy for managing functional gastrointestinal symptoms in IBD, particularly during disease flares. We have expanded the Discussion section to provide a more detailed analysis of the low FODMAP diet, emphasizing its clinical benefits and distinguishing it from general food avoidance behaviors. We have also included relevant references to support this discussion (see page 9, paragraph 4, line 355).
Comment 2:
Another critical omission is the discussion of food-based medical therapies, such as exclusive enteral nutrition (EEN) with formulations like Modulen or the elemental diet for Crohn’s disease. These are well-established interventions, particularly in pediatric populations, but also have applications in adults who require nutritional optimization without exacerbating inflammation. Given the review's focus on knowledge gaps, it would be beneficial to assess whether patients are aware of these therapeutic options and their clinical efficacy.
Response 2:
Thank you for raising this important point. We have now included a dedicated section in the Discussion addressing food-based medical therapies, such as exclusive enteral nutrition (EEN) and elemental diets. We also discuss the knowledge gaps surrounding these therapeutic options among patients with IBD and provide updated references to support our analysis (see page 8, paragraph 4, line 346).
Comment 3:
Overall, this is a well-researched and thoughtful review that highlights an often-overlooked aspect of IBD management. Addressing the above points would further strengthen its clinical relevance and applicability.
Response 3:
We sincerely appreciate your positive feedback. We believe that the revisions made in response to your valuable suggestions have further enhanced the clarity, clinical relevance, and applicability of our manuscript.
Reviewer 2 Report
Comments and Suggestions for Authors
Addressing Nutritional Knowledge Gaps in Inflammatory Bowel Disease: A Scoping Review
The manuscript provides an insightful and well-organized review of the nutritional knowledge gaps among individuals with inflammatory bowel diseases (IBD). The authors effectively highlight the critical role nutrition plays in managing IBD symptoms, emphasizing the need for tailored educational interventions to address these gaps. The research methodology is robust, utilizing the Arksey and O'Malley framework and adhering to PRISMA 2020 guidelines, ensuring transparency and rigor in the study selection process. This article presents valuable findings that can guide future interventions aimed at improving the dietary management of IBD patients.
Specific Suggestions:
- The manuscript should better define what constitutes "nutritional knowledge" for IBD patients. Are the authors referring solely to specific food choices, or are broader aspects like meal planning and nutrient composition included? A clearer definition would benefit the reader. While the paper touches on common misconceptions among IBD patients, it would be helpful to include more detailed discussion on the existing evidence-based nutritional guidelines for managing IBD. Are there any established best practices or guidelines the article could refer to?
- The conclusion highlights the need for future research into evidence-based interventions for IBD patients. It would be helpful if the manuscript could specify areas of research that are particularly lacking, such as the role of specific dietary patterns (e.g., Mediterranean diet) or the impact of timing and frequency of interventions.
- Since dietary practices are heavily influenced by cultural beliefs, the paper could benefit from a more in-depth discussion of how cultural differences may affect nutritional knowledge and practices among IBD patients. Are there any studies addressing this aspect?
- While there is mention of micronutrient deficiencies, such as calcium and zinc, in patients who avoid certain foods, a more comprehensive breakdown of common nutritional deficiencies in IBD patients would be useful. For instance, highlighting deficiencies in specific vitamins and minerals would provide further clarity on the importance of dietary interventions. The review could further explore how healthcare providers can integrate dietary counseling into routine care. For example, should dietitians be more involved in IBD treatment teams, and if so, what role can they play in improving patient outcomes?

Author Response
Comment 1:
The manuscript should better define what constitutes "nutritional knowledge" for IBD patients. Are the authors referring solely to specific food choices, or are broader aspects like meal planning and nutrient composition included? A clearer definition would benefit the reader.
Response 1:
Thank you for your observation. We have clarified the definition of "nutritional knowledge" in the Introduction section. (see page 2, paragraph 1, line 70).
Comment 2:
While the paper touches on common misconceptions among IBD patients, it would be helpful to include more detailed discussion on the existing evidence-based nutritional guidelines for managing IBD. Are there any established best practices or guidelines the article could refer to?
Response 2:
We appreciate your suggestion. We have expanded the Discussion section to provide a more detailed overview of current evidence-based nutritional guidelines for managing IBD, including the ESPEN and ECCO guidelines (see page 8, paragraph 4, line 308).
Comment 3:
The conclusion highlights the need for future research into evidence-based interventions for IBD patients. It would be helpful if the manuscript could specify areas of research that are particularly lacking, such as the role of specific dietary patterns (e.g., Mediterranean diet) or the impact of timing and frequency of interventions.
Response 3:
Thank you for this insightful recommendation. We have revised the Conclusion section to specify areas of research that remain underexplored, such as the role of specific dietary patterns (e.g., Mediterranean and low FODMAP diets) and the impact of the timing and frequency of dietary interventions on disease outcomes (see page 9-10, paragraph 5, line 338).
Comment 4:
Since dietary practices are heavily influenced by cultural beliefs, the paper could benefit from a more in-depth discussion of how cultural differences may affect nutritional knowledge and practices among IBD patients. Are there any studies addressing this aspect?
Response 4:
We agree with your observation and have expanded the Discussion section to explore how cultural beliefs influence nutritional knowledge and practices among IBD patients. W (see page 9, paragraph 4, line 378).
Comment 5:
While there is mention of micronutrient deficiencies, such as calcium and zinc, in patients who avoid certain foods, a more comprehensive breakdown of common nutritional deficiencies in IBD patients would be useful. For instance, highlighting deficiencies in specific vitamins and minerals would provide further clarity on the importance of dietary interventions.
Response 5:
Thank you for your valuable suggestion. We have revised the Discussion sections to provide a more comprehensive breakdown of common nutritional deficiencies among IBD patients, including vitamins (e.g., D, B12, folate) and minerals (e.g., iron, calcium, zinc). This detailed overview underscores the importance of appropriate dietary interventions (see page 7, paragraph 4, line 273).
Comment 6:
The review could further explore how healthcare providers can integrate dietary counseling into routine care. For example, should dietitians be more involved in IBD treatment teams, and if so, what role can they play in improving patient outcomes?
Response 6:
We completely agree with your observation. In the Discussion section, we have added a dedicated paragraph emphasizing the importance of integrating dietitians into IBD treatment teams. (see page 9, paragraph 4, line 366).